# Addressing Sample Complexity in Visual Tasks Using Hindsight Experience Replay and Hallucinatory GANs

**Himanshu Sahni** [1 2]  **Toby Buckley** [1]  **Pieter Abbeel** [1 3]  **Ilya Kuzovkin** [1]

## Abstract

Reinforcement Learning (RL) algorithms typically require millions of environment interactions to learn successful policies in sparse reward settings. Hindsight Experience Replay (HER) was introduced as a technique to increase sample efficiency by re-imagining unsuccessful trajectories as successful ones by changing the originally intended goals. However, HER cannot be directly applied to visual environments where goal states are characterized by the presence of distinct visual features. In this work, we show how visual trajectories can be hallucinated to appear successful by altering agent observations using a generative model trained on relatively few snapshots of the goal. We then use this model in combination with HER to train RL agents in visual settings. We validate our approach on 3D navigation tasks and a simulated robotics application and show marked improvement over standard RL algorithms and baselines derived from previous work.

## 1. Introduction

Deep Reinforcement Learning (RL) has recently demonstrated success in a range of previously unsolved tasks, from playing Atari and Go on a superhuman level (Mnih et al., 2015; Silver et al., 2017) to learning control policies for real robotics tasks (Levine et al., 2016; OpenAI, 2018; Pinto et al., 2017). But deep RL algorithms are highly sample inefficient for complex tasks and learning from sparse rewards can be challenging. In these settings, millions of steps are wasted exploring trajectories that yield no learning signal. On the other hand, providing dense rewards along these trajectories is a tedious job that requires substantial domain knowledge and RL expertise. Shaping the

[1]**Offworld Inc.** [2]**Georgia Institute of Technology** [3]**University of California Berkeley**. Correspondence to: Himanshu Sahni <hsahni3@gatech.edu>.

*Reinforcement Learning for Real Life (RL4RealLife) Workshop in the 36th International Conference on Machine Learning*, Long Beach, California, USA, 2019. Copyright 2019 by the author(s).

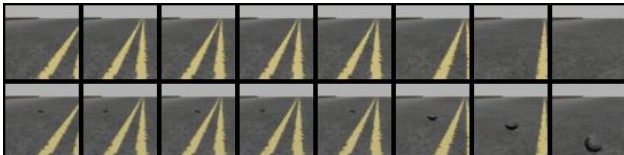

Figure 1. VHER works by using a generative model to hallucinate the presence of goals at the end of unsuccessful trajectories. The agent's task is to search for a pebble randomly placed in its surroundings and collect it by approaching and centering it in its view. The top row shows a failed trajectory which ends in the agent not finding the pebble. The bottom row replays the same trajectory with a hallucinated visual goal inserted by HALGAN at every state such that a pebble appears to be collected.

rewards in an attempt to make learning easier is non-trivial and can often lead to unexpected hacking behaviour (Ng et al., 1999; Randløv & Alstrøm, 1998). Therefore, an important vector for RL research is towards more sample efficient methods that minimize the number of environment interactions, yet can be trained using only sparse rewards. To this end, Andrychowicz et al. (2017) introduced Hindsight Experience Replay (HER), which can rapidly train a goal-conditioned policy by retroactively imagining failed trajectories as successful ones. By making use of failed attempts to increase sample efficiency, HER was able to learn a range of robotics tasks that traditional RL approaches are unable to solve. But HER was only shown to work in non-visual environments, where the precise goal configuration is provided to the agent's policy throughout training and where it is straightforward to find a goal that is satisfied in any state. It also relied on the use of universal value function approximators (UVFAs) (Schaul et al., 2015) to generalize over multiple goals. Thus, it is not directly applicable to challenging visual domains where the agent's observations must be altered in order to change the goal and where the goal location is not explicitly known a priori and must be searched for within the environment.

Yet, we desire for RL agents to quickly learn to operate in the uncertain visual environments that humans inhabit. Some recent work has extended HER to visual domains where goals are sampled from the set of possible agent states and provided to the agent as an input (Nair et al., 2018). But there is a wide range of visual tasks where we do not have

an explicit representation of a goal beforehand and where a failed state may not easily map to a goal state. We would like the agent to be able to perform these tasks without providing it a specification of the goal during execution and instead have it search for the goal in its environment. For this, the agent must be able to infer the presence of goals from the state image itself, automatically learning to generalize over multiple goal configurations.

To address high sample complexity of RL in such visual environments, we introduce a new algorithm for visual hindsight experience replay, which combines a hallucinatory generative model (HALGAN) with HER to rapidly solve tasks using only raw pixels in the state as input to the agent policy. HALGAN minimally alters images in snippets of failed trajectories to appear as if the desired goal is achieved at the end. In order to retroactively hallucinate success in a visual environment, it is necessary to alter the state images along the entire failed trajectory to make it appear as if the goal was present throughout (see figure 1). HALGAN is trained using a few snapshots of *near goal* images, where the relative location of the agent to the goal is known. It is then combined with HER during reinforcement learning, where goal location is unknown, to hallucinate goals along unsuccessful trajectories. We make use of the assumption that in realistic robotic applications, while it may be difficult to obtain the explicit location of the goal throughout reinforcement learning, one can obtain the configuration of the robot relative to itself easily. This can be done using SLAM or other state tracking techniques (Montemerlo et al., 2002). We will primarily focus on tasks where the completion of a goal can be visually identified within the agent state.

The key contributions of this work are to expand the applicability of HER to visual domains by providing a way to retroactively transform failed visual trajectories into successful ones and hence allowing the agent to rapidly generalize across multiple goals using only the state as input to its policy. We aim to do so in conjunction with minimizing the amount of direct goal configuration information required to train HALGAN. We believe that the sample complexity reduction HALGAN provides is an important step towards being able to train RL policies directly in the real world.

## 2. Background

**Reinforcement Learning.** In reinforcement learning, the agent is tasked with the maximization of some notion of a long term expected reward (Sutton & Barto, 2018). The problem is typically modeled as a Markov decision process (MDP). An MDP consists of a tuple $< S, A, R, T, \gamma >$, where $S$ is the set of states the agent can exist in, $A$ is the set of environment actions, $R : S \times A \to \mathbb{R}$ is the function mapping states and actions to a scalar reward, $T : S \times A \to S$ is the transition function, and $\gamma \in [0, 1)$ is a discount

factor that weighs the importance of future rewards versus immediate ones. Stochasticity in the environment can be present in the form of uncertainties in transition or reward.

The agent must learn a policy, $\pi : S \to A$, mapping every state to an action. The optimal policy, $\pi^*$, is often the goal of learning. It informs the agent on an action that typically maximizes expected value of the sum of future discounted rewards, $\mathbb{E}[\sum_k \gamma^k R(s_{t+k})]$, starting from any state $s_t$. This expectation, known as the state value ($V : S \to \mathbb{R}$), is over trajectories experienced under the current policy and environment dynamics. UVFAs (Schaul et al., 2015) approximate value functions with respect to a goal in addition to the state, $V : S \times G \to \mathbb{R}$. The optimal policy, $\pi^*(s; g)$, in this case maximizes the probability of achieving a particular goal, $g$, from any state.

Off-policy RL algorithms can learn an optimal policy using experiences from a *behavior policy* separate from the optimal policy. In particular, off-policy algorithms can make use of samples collected in the past, leading to more sample efficient learning. An experience replay (Lin, 1992) is typically employed to store past transitions as tuples of $(s_t, a_t, r_t, s_{t+1})$. At every step of training, a minibatch of transitions is sampled from the replay at random and a loss on future expected return minimized. The off-policy algorithms employing an experience replay used in this work are Double Deep Q-Networks (DDQN) (Van Hasselt et al., 2016) and Deep Deterministic Policy Gradients (DDPG) (Lillicrap et al., 2015).

**Hindsight Experience Replay.** HER was shown to achieve speedups in learning in environments where the goal configuration is provided along with the agent state to the policy. The essential idea is to store each trajectory, $Traj_i = s_0^i, s_1^i, ..., s_T^i$, with a number of additional goals along with the originally specified one. An off-policy algorithm employing an experience replay is used to train a UVFA which learns a policy which generalizes across multiple goals. During replay, the original goals are changed to states that have actually been achieved by the agent in the past.

The reward is also modified retroactively to reflect the new goal being replayed. In particular, HER assumes that every goal, $g \in G$, can be expressed as a predicate $f_g : S \to \{0, 1\}$. That is to say, all states can be judged as to whether or not a goal $g$ has been achieved in them. Thus, while replaying the trajectory $Traj_i$ with a surrogate goal $\overline{g}$, one can easily reassign rewards along the entire trajectory as,

$$r_{\overline{g}}(s_t^i) = \begin{cases} 1 & if f_{\overline{g}}(s_t^i) = 1 \\ 0 & otherwise. \end{cases}$$

Andrychowicz et al. (2017) report that selecting $\overline{g}$ to be a future state from within the same (failed) episode leads to the best results. This training approach forms a sort of implicit curriculum for the agent. In the beginning, it encourages the

agent to explore further outwards along trajectories it has visited before. Since the surrogate goal, $\overline{g}$, is also explicitly provided to the UVFA policy, it soon learns to also generalize this curriculum over unseen goals. Over time, the agent is able to achieve any goal in $G$, including the real ones.

**Wasserstein GANs.** We employ an improved Wasserstein ACGAN (Gulrajani et al., 2017; Odena et al., 2017) as our generative model because of its stability, realistic looking outputs, and ability to condition the generated images on a desired class. A typical W-ACGAN has a generator, $H$, that takes as input a class variable and a latent vector of random noise. It then generates an image which is fed into the discriminator, $D$. $D$ rates the image on its fidelity to the training data and, as an auxiliary task, predicts class membership. The Wasserstein distance between the distributions of real, $p_R$, and generated, $p_H$, images is used as a loss to train the combined model. A GAN enables us to produce realistic looking hallucinations that will allow the agent to easily generalize from imagined goals to real ones. Realistic insertion of goals was not an issue in HER because a new goal could directly be substituted in a replayed transition without any modification to the states.

## 3. Related Work

**Generative Models in RL.** In recent years, generative models have demonstrated significant improvements in the areas of image generation, data compression, denoising, and latent-space representations, among others (Goodfellow et al., 2014; Chen et al., 2016; Vincent et al., 2008). Reinforcement learning has also benefited from incorporating generative models in the training process. Ha & Schmidhuber (2018) synthesize a lot of prior work in the area by proposing a Recurrent Neural Network (RNN) based generative dynamics model (Schmidhuber, 1990) of popular OpenAI gym (Brockman et al., 2016) and VizDoom (Kempka et al., 2016) environments. They employ a fairly common procedure of encoding high dimensional visual inputs from the environment into lower dimension embedding vectors using a Variational Auto Encoder (VAE) (Kingma & Welling, 2013) before passing it on to the RNN model. Held et al. (2017) use a GAN to generate goals matching in difficulty to an agent's skill on a task. Called GoalGAN, it generates an automatic curriculum of incrementally harder to reach goals. But it assumes that goals can easily be set in the environment by the agent and does not make efficient use of trajectories that failed to achieve these objectives. Generative models have also been used in the closely related field of imitation learning to learn from human demonstrations or observation sequences (Ho & Ermon, 2016; Edwards et al., 2018b; Schroecker et al., 2019). In our approach, we do not require demonstrations of the task, or even a sequence of observations, but relatively few random snapshots of the

goal with a known configuration which we use to speed up reinforcement learning.

**Goal Based RL.** Some recent work has focused on leveraging information on the goal or surrounding states to speed up reinforcement learning. Edwards et al. (2018a) and Goyal et al. (2018) learn a reverse dynamics model to generate states backwards from the goal which are then added to the agent's replay buffer. The former assumes that the goal configuration is known and backtracks from there, whereas in the latter, high-value states are picked from the replay buffer or a GoalGAN is used to generate goals. The latter work also learns an inverse policy, $\pi(a_t|s_{t+1})$ to generate plausible actions leading back from goal states. In contrast, we focus on minimally altering states in past failed trajectories to appear as if a goal has been completed in them. This avoids having to generate entirely new trajectories and allows us to make full use of the environment dynamics already present in previous state transitions.

Others have focused on learning goal-conditioned policies in visual domains using a single or few images of the goal (Xie et al., 2018; Zhu et al., 2017). Nair et al. (2018) train a $\beta$-VAE (Burgess et al., 2018) on state images for a threefold purpose: (1) to sample new goals during training, (2) to use the Euclidean distance between feature encodings of current and goal images as a dense reward, and (3) to retroactively alter goals with VAE generated images and reassign rewards appropriately. The set of goals $G$ is assumed to be the same as the set of states $S$ and hence they are easy to swap back and forth. This works well for domains where the goal is separately provided to the policy along with the agent state, and where states do not have to be modified for changing goals. In this work, we attempt learning in domains where the goal image is not known beforehand and thus cannot be provided to the agent's policy, and where the goal may or may not be present in a particular agent state.

## 4. The missing component in HER

First, we will more formally discuss what is missing from the original HER formulation that does not allow it to readily extend to visual domains. Then, in the next section, we will describe in detail how the use of hallucinatory generative models can help bridge the gap.

HER makes an assumption that "given a state $s$ we can easily find a goal $g$ which is satisfied in this state" (Andrychowicz et al., 2017). It requires a mapping, $m : S \to G$ that maps every state $s \in S$ to a goal $g \in G$ that is achieved in that state. While this mapping may be relatively straightforward to hand design for real-valued state spaces, its analog for visual states cannot be constructed easily. For example, if the state space of the agent lies on the plane of real values in $\mathbb{R}^2$, the goal may be to achieve a particular $x$-coordinate. So

in the agent state $(x = 0.5, y = 1.0)$, a goal that is satisfied is simply $g : x = 0.5$. Now imagine if the agent must instead navigate to a beacon on a 2D plane using camera images as state inputs. In order to convert any arbitrary state into one in which a goal is satisfied, the beacon must be visually inserted into the image itself. We call these *goal hallucinations* (see figure 2).

In order to fully utilize the power of HER, not only should the agent be able to hallucinate goals in arbitrary states, but also consistently in the same absolute position throughout the failed trajectory. Note that with each step along the trajectory, the position of the goal (the beacon) changes relative to the agent's and thus the agent's observation must be correctly updated to reflect this change. The goal must *appear* to have been solved in a future state along every step of the trajectory. Only then can we make use of the existing transitions along the entire trajectory for replay with hallucinated as well as original goals. Thus, visual settings require the mapping $m$ to be extended along the entire trajectory $s_0, \ldots, s_T \in S_{Traj}$; $m_V : S_{Traj}^T \rightarrow G$, where $T$ is the maximum length of a trajectory and $Traj$ is the space of failed trajectories. Every state $s$ along a trajectory from $Traj$ must be modified by the mapping into a near-goal state that is consistent with the final goal state of that trajectory. This work's main contribution lies in showing that such a mapping can be learned by a generative model using some knowledge of the goal in the form of *goal snapshots* with known relative location. We show how the learned model mapping unsuccessful trajectories to successful ones can be applied to training RL agents whose policy is solely conditioned on their state image.

# 5. Approach

To address the shortcomings of HER in visual domains, we adopt a two-part approach. First, a generative network, HALGAN, is trained to modify any existing state from a failed trajectory into a goal or *near goal* state. HALGAN generates goal hallucinations conditioned on the configuration of the robot in the current state relative to its own configuration in a future state from the same episode. Note that the choice to condition goal generation on relative location was made in light of the focus on sample efficiency.

During reinforcement learning, random snippets of failed trajectories experienced in the past are replayed with the final state in the snippet designated as the target goal location. The trained HALGAN uses the configuration of the agent relative to itself in the future end state to modify a pair of states to appear as if a goal was indeed achieved during this trajectory. Details of the entire hallucinating process are provided in the next few subsections.

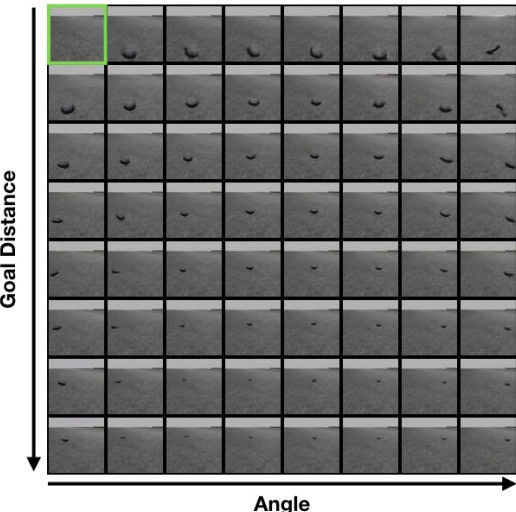

*Figure 2.* Hallucinated images generated by our model. The original, failed, image is on the top left. All others are including goals generated by HALGAN. The goal distance is increased from top to bottom and angle from left to right. This image demonstrates that using our training approach, goal hallucinations can be generated with high fidelity in any relative configuration.

## 5.1. Hallucinating Visual Goals

First, HALGAN is trained on a dataset, $R$, of observations of the goal where its relative location to the agent is explicitly known. These snapshots of the goal can be collected beforehand and are only used once to train the generative model. Then, it generalizes to create thousands of hallucinations along failed trajectories during reinforcement learning using only agent configuration. These failed trajectories are ones the agent has taken in the past and are stored in its experience replay.

In order to "fool" the agent into thinking that it has indeed achieved a goal, one has to insert the goal into every image of the trajectory. Thus, the states $s_0, s_1, ..., s_T$ have to be modified to $\overline{s_0}, \overline{s_1}, ..., \overline{s_T}$ such that it appears as if the goal has been achieved in $\overline{s_T}$. This is in contrast to the regular HER approach, or the approach by Nair et al. (2018), where the state can be directly mapped to a goal using a hand designed mapping. The hallucinated goal location must remain consistent throughout the replayed trajectory so as to not appear to violate the environment's dynamics. In the following subsections, we describe each component of HALGAN and then show how it fits together to generate consistent hallucinations of the goal.

## 5.2. Minimal Hallucinations

One of our aims is to minimally alter a failed trajectory in order to turn its states into goal ($\overline{s_T}$) or *near-goal* ($\overline{s_0}, \overline{s_1}, \ldots, \overline{s_{T-1}}$) states. This makes full use of existing trajectories and does not require HALGAN to re-imagine

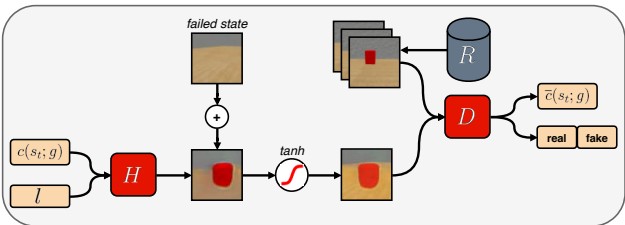

*Figure 3.* A conditioning vector $c(s_t; g)$ informs the generator, $H$ on the desired relative location of the goal. $l$ is a random noise vector drawn from $\mathcal{N}(1, 0.1)$. The generated goal image is added to a *failed state* and then passed through a renormalizing $tanh$ function. This is the final hallucinated state with the goal positioned as desired. $H$ is trained adversarially along with $D$, which is learning to rate the fakes and real near goal images from the dataset $R$. $D$ also predicts relative goal configurations in real and fake images, which in turn incentivizes $H$ to hallucinate goals in the correct relative locations.

the environment dynamics or unnecessary details about the goal state such as the background.

To this end, we train an additive model, such that the generator, $H$, has to produce only differences to the state image that add in the goal. To obtain a hallucinated image $\overline{s_t}$ with the goal at the final state of the trajectory, $s_T$, we compute,

$$\overline{s_t} = Tanh\left(s_t + H\left(c(s_t; s_T), l\right)\right), \quad (1)$$

where, $H$ is the generative model function, $c(s_t; g)$ is the relative configuration of the robot to a desired goal state $g$ and $l$ is a random latent conditioning vector. $Tanh$ is used to re-normalize the hallucinated state image to $[-1, 1]$. Any differentiable bounded function can be used for this purpose. The hallucinated state, $\overline{s_t}$, along with a state $s_r$ sampled from dataset $R$, is then fed to the discriminator $D$ to compute the discriminative loss,

$$L_D = \mathbb{E}_{\overline{s_t} \sim p_H}[log D(\overline{s_t})] - \mathbb{E}_{s_r \sim p_R}[log(D(s_r))]. \quad (2)$$

In addition to the discriminator loss, a gradient penalty is employed in the improved training of Wasserstein GANs (see Gulrajani et al. (2017) for more details).

$$L_\nabla = \mathbb{E}_{\hat{s} \sim P_{\hat{s}}}\left(\|\nabla D(\hat{s})\|_2 - 1\right)^2 \quad (3)$$

As a result of generating only image differences, the trained hallucinatory model is invariant to certain visual variations, such as background, presence of other objects, etc.

To encourage the model to generate minimal modifications to the original failed image, we also add a $L_2$ norm loss on the output of $H$. In our experiments, this helped in discouraging the generator from focusing on unnecessary elements of goals such as background information or extra

objects in the environment.

$$L_H = \|H\left(c(s_t; s_T), l\right)\|_2 \quad (4)$$

### 5.3. Regression Auxiliary Task

Typical ACGANs are conditioned on a discrete set of classes, such as flower, dog, etc (Odena et al., 2017). In our approach, the generator is conditioned on the relative configuration of the agent from the desired goal state, which is a real-valued vector $c(s_t; g) \in \mathbb{R}^n$. The auxiliary task for the discriminator is to regress to the real valued relative location of the goal seen in a training image. To train this regression based auxiliary task, we use a *mean squared error* loss,

$$L_A = \|\overline{c}(s_t; g) - c(s_t; g)\|_2 \quad (5)$$

where $\overline{c}(s_t; g)$ is the relative configuration predicted by $D$. We found it helpful to add a small amount of Gaussian noise to our auxiliary inputs for robust training, especially on smaller datasets. We also found drawing the latent conditioning vector $l$ from a normal distribution centered around 1 lead to more accurate auxiliary regression.

### 5.4. HALGAN

Our final loss to the combined HALGAN is,

$$L = L_D + \alpha L_\nabla + \beta L_H + \lambda L_A \quad (6)$$

where, $\alpha$, $\beta$, and $\lambda$ are weighting hyperparameters, which we set to 10, 1, and 10 respectively in all our experiments.

To summarize, the training process is as follows. The generator, conditioned on a randomly drawn relative goal location produces a difference image which is then added to a randomly selected image from a failed trajectory to create a goal hallucination. The discriminator is provided with these hallucinated images as well as ground truth images from $R$ and has to score the images on their authenticity and also predict the relative goal location. See figure 3 for a representation of the HALGAN training process and the appendix for more details on the network architectures and training procedure.

For the purposes of our experiments, we collect the training data for HALGAN, $R$, by using the last few states of a successful rollout. Note that the exact data required in $R$ are randomly selected snapshots from near the goal with known relative poses. Since at the end of a successful rollout, agent configuration corresponds to goal configuration, relative configuration for training HALGAN can be computed using only agent configuration information. Only observations, including the state image and agent configuration, are used, no actions have to be provided or demonstrated. This alleviates the data collection burden as the human does not have to demonstrate the optimal completion of the task and

snapshots can be collected in any order. For example, it is significantly simpler to record the desired final configuration of objects on a table than to record a full, optimal demonstration of a robot arm aranging them. It also allows the generative model to be independent of the agent and demonstrator action spaces. We also collect a dataset of failed trajectories using random exploration. These are to train HALGAN by adding to the output of $H$ and creating hallucinated *near goal* states. Most off-policy RL methods that employ an experience replay have a *replay warmup* period where actions are taken randomly to fill the replay to a minimum before training begins. This dataset of failed trajectories can be the same as the replay warmup and no extra exploration is required.

## 5.5. Visual HER

During reinforcement learning, the agent explores its environment as normal. Every time a batch is sampled for training, a few of the data points from it are augmented with goal hallucinations. The detailed process is explained in algorithm 1. The result is that the agent encounters hallucinated near goal states with a much higher frequency than if it were randomly exploring. This in turn encourages the agent to explore further from *near goal* states.

---

**Algorithm 1** Visual Hindsight Experience Replay

1: **Given:** Trained hallucinatory model $H$, Reward reassignment strategy $r_g(s)$.
2: Initialize off-policy Algorithm $\mathbb{A}$. {eg. DDQN, DDPG}
3: Initialize Experience Replay $E$ by random exploration.
4: **for** step$= 1, N$ **do**
5:     Sample an action according to behavior policy $a_t \leftarrow \pi(s_t)$ in current state.
6:     Execute $a_t$ in the environment and observe state $s_{t+1}$, reward $r_t$.
7:     Store tuple $\langle s_t, a_t, r_t, s_{t+1} \rangle$ in $E$.
8:     Sample minibatch $B$ from $E$ for training.
9:     **for** $e = \langle s_i, a_i, r_i, s_{i+1} \rangle$ in $B$ **do**
10:         Sample $c \sim Bern(p)$ {$p =$ hallucination prob.}
11:         **if** $c$ **then**
12:             Sample $d \sim Unif(\{0, 1, ..., D\})$ {distance to goal state}
13:             Compute relative configurations $c(s_i; s_{i+d})$ and $c(s_{i+1}; s_{i+d})$. {Setting $s_{i+d}$ as the goal state}
14:             $s_i \leftarrow s_i + H(c(s_{i+d}; s_i), l)$
15:             $s_{i+1} \leftarrow s_{i+1} + H(c(s_{i+d}; s_{i+1}), l)$
16:             $r_i \leftarrow r_{s_{i+d}}(s_{i+1})$
17:         **end if**
18:     **end for**
19:     Perform one step of optimization using $\mathbb{A}$ on the modified minibatch $B$.
20: **end for**

---

An important consideration is the retroactive reassignment of rewards. As a reminder, HER uses a manually defined function $f_g(s)$ which decides if the goal $g$ is satisfied in a state $s$ to designate rewards during hindsight replay. This sort of retroactive reward function is hard to hand design in visual environments. Comparing state and goal images pixel by pixel is typically ineffective. Fortunately, for the purposes of reward reassignment during hindsight replay, one need only compare the agent state to a future one in the same episode. Hence, a similar function, $f_s : S \times S \rightarrow \{0, 1\}$, which decides whether a pair of states are the same for the purpose of goal completion, can also be used to reassign rewards. As mentioned in section 3, Nair et al. (2018) use a trained $\beta$-VAE as $f_s$ to reassign rewards in a dense manner. Here, we make use of the access to the robot's own configuration to design a similar function, $f_c$, where $c$ is the robot configuration at a particular state. We then assume that any goal satisfied in $c$ must also be satisfied in any other state with a similar enough configuration. During retroactive reward reassignment, we compare the relative configurations of the agent in the current and final state, and hallucinate a reward if they are the same.

## 6. Experiments

We test our method on two first person visual environments. In a modified version of MiniWorld (Chevalier-Boisvert, 2018), we design two tasks. The first one is to *navigate* to a red box located in an enclosed room (figure 4 left). The second task is to *successively navigate*, first to the red box, picking it up by visually centering it, and then carrying it to a green box somewhere else in the room (see figure 4 center).

The second environment is a more visually realistic simulated robotics domain, where a TurtleBot2 (Wise & Foote) equipped with an RGB camera is simulated within Gazebo (Koenig & Howard, 2004). We use gym-gazebo (Zamora et al., 2016) to interface with Gazebo. In this environment, the agent must collect a pebble scattered randomly on a road by approaching and centering it in its visual field (figure 4 right). The episode ends and the agent is reset to the starting location if it wanders too far.

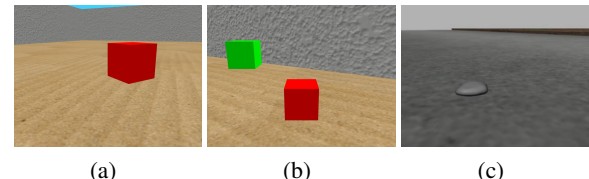

     (a)            (b)            (c)

*Figure 4.* Example of a *near goal* state in Turtlebot (left) and MiniWorld navigate (center) and pick-and-place (right) environments.

Figure 4 depicts near goal states in all of our tasks. The

goal is randomly spawned a small distance away from the agent. The environment only provides a sparse reward of 1 for achieving the goal. No reward shaping is done for either the RL baselines or our approach. Encountering the goal during exploration is extremely rare and standard RL is sample inefficient or completely ineffective. The size of the near goal dataset, $R$, for the Turtlebot, *navigation* and *successive navigation* tasks is 6840, 2000, and 6419 images with relative goal configurations respectively. Though, we show that the effect of reduction in the amount of *near goal* states leads to little performance degradation in the Turtlebot environment (figure 6).

In the Turtlebot and MiniWorld *navigation* tasks, the configuration of the agent is simply it's $\langle x, y, yaw \rangle$. In *successive navigation*, an additional binary field indicates whether the red box is held by the agent. The agent's relative configuration is calculated with respect to the red box before it is picked up, and the green box afterwards. Hallucinations are generated for the agent approaching both boxes. Hallucinated reward issued by HALGAN is also sparse, +1 only if the hallucinated state happens to be a goal state. We found it helpful to anneal the amount of hallucinations in a batch over time as the agent starts filling the replay with real reward. Details of the annealing rate and other experimental hyperparameters are provided in the appendix.

**Comparisons.** There is no prior work that attempts HER in visual domains without explicit goal conditioning. Hence, we compare our approach to multiple extensions of a close existing approach and standard RL baselines. First, a naive extension of HER into the visual domain, *her*, simply rewards the agent for states at the end of failed trajectories during replay without hallucinating. Hence, the agent receives hindsight rewards, but the sampled trajectories still seem to end in failures. This tests out the approach without the presence of HALGAN to establish a baseline without hallucinations.

A second baseline is derived from Nair et al. (2018)'s work (*RIG*) in training goal-conditioned policies with a dense reward based on the distance between the embedding of the sampled state and that of a goal image. *RIG*'s retroactive reassignment of goals relies on the use of UVFAs, which is not possible for our domains where the goal image is unknown. Therefore, we test two variants of this baseline where we attempt to find a suitable comparison. We first train a VAE on data available to HALGAN, i.e. *near goal* images in $R$ and failed state images collected by random exploration. Then, during RL, *vae-her* simply sets the final image in a failed trajectory, without any hallucinations, as the goal and uses the trained VAE to compute reward for a transition along that trajectory. This baseline evaluates the effectiveness of dense reward shaping in our domains without the use of hallucinations from HALGAN.

Next, *rig-* follows a similar dense reward shaping strategy, but computes distance of a state to a randomly sampled goal image in $R$, filtered for using the agent's relative configuration. Hence, *rig-* rewards the agent for being in states that look similar to goal states in retrospect. Note that these extensions to *RIG* only do dense reward shaping and no goal reassignment as *RIG* is only designed for goal conditioned tasks. As we show in the results, using HALGAN to hallucinate goals allows us to train better agents while keeping the rewards sparse. All baselines used the exact same hyperparameters as our approach.

For the the distance based rewards provided by the VAE in *rig-* to be the same order of magnitude as the environment rewards, it was necessary to re-scale them. The scaling factor in all our experiments was set to $0.02$.

**Discrete and Continuous Control.** An advantage of our method is that HALGAN is agnostic to the agent's action space as a result of directly conditioning on the relative location of the robot to a state in the future. Hence we can apply our method easily to both discrete and continuous control tasks. In the discrete TurtleBot environment, the action space is back and forth movement and turning (4 actions). The base off-policy algorithm used is Double DQN (Van Hasselt et al., 2016). For the continuous MiniWorld environments, actions are the linear and rotational velocities of the agent at the next step, capped at a fixed amount. A penalty on the $L_2$ norm of the output actions is applied at each step to simulate energy step cost. Otherwise, the agent is only provided the sparse task completion reward. The base algorithm used in this setting is DDPG (Lillicrap et al., 2015). We employ deep convolutional neural networks as function approximators that take in the state image as input and outputs the desired control actions or values.

## 7. Results

In all of our experiments, HALGAN trained agent begins learning immediately (figure 5). This is due to the realistic looking hallucinated goals being quickly identified as desirable states. This is in contrast to standard RL which rarely encounters reward and must explore at length to encounter random rewards in order to begin learning, if at all.

In the discrete TurtleBot pebble collection domain (figure 5 left), the naive HER strategy provides a good enough exploration bonus for the agent to explore further and quicker than standard DDQN. It begins learning by 100K steps. HALGAN agent, by contrast, starts learning to navigate to real goals immediately.

For the continuous control experiments in MiniWorld (figure 5 center, right), only HALGAN agent is able to learn to complete the task. Note that achieving a reward of $0$ in this environment is relatively easy, it is only positive rewards

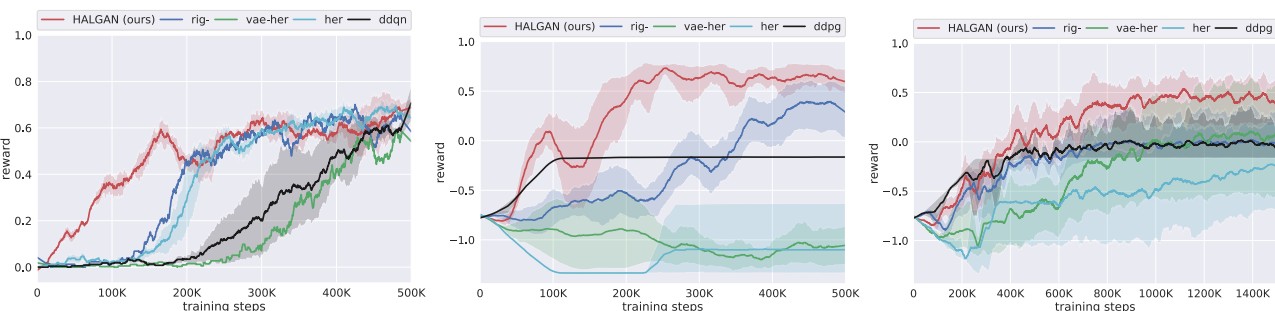

*Figure 5.* In all tasks, VHER starts learning immediately whereas the baselines needs to explore far more to randomly encounter positive rewards. In the Turtlebot pebble collection task (left), all algorithms eventually learn an optimal policy but VHER begins learning immediately and converges quickly. In the harder, continuous control MiniWorld *navigate* task (middle), neither DDPG nor naive-HER are able to learn to complete the task. Only the *rig-* baseline somewhat learns the task eventually on three of the five random seeds. In the final *pick-and-place* task, only VHER learns the optimal policy in four out of five random seeds.

that indicate achievement of goal. DDPG never encounters any reward during exploration and hence learns to simply minimize its actions in order to avoid movement penalty. Naive *her* initially encourages exploration and hence incurs a heavy penalty, but doesn't learn to associate the hallucinated rewards it receives with the presence of a goal. Some of the random seeds eventually converge to the same degenerate policy as DDPG. *vae-her*, the augmentation of *her* with dense rewards from a trained VAE, also proves unsuccessful for either task, demonstrating that dense rewards without hallucinated or real goals in failed trajectories are also ineffective for learning in these domains. Only the *rig-*strategy of providing dense rewards relative to random goal images eventually learns to complete the *navigation* task for some of the seeds. For the *successive navigation* task, *rig-* only learns a working policy on a single seed and the other baselines perform similarily or worse. Interestingly, *rig's* dense reward reassignment can be readily combined with our approach of state modification by hallucination, providing directions for future work.

Finally in figure 6, we show the change in performance on the TurtleBot pebble collection task due to using fewer training samples in $R$. The effect is only slightly slower learning even for the largely reduced dataset of only 1000 images. The minimalistic hallucinations created by HALGAN require a relatively small amount of data to train well enough to provide a significant boost in reinforcement learning.

## 8. Discussion

A major impediment to training RL agents in the real world is the amount of data an agent must collect before it can start drawing inference on which actions lead to rewards and which ones are to be avoided. High sample complexity makes problems such as fragility of physical systems, energy consumption, speed of robots and sensor errors man-

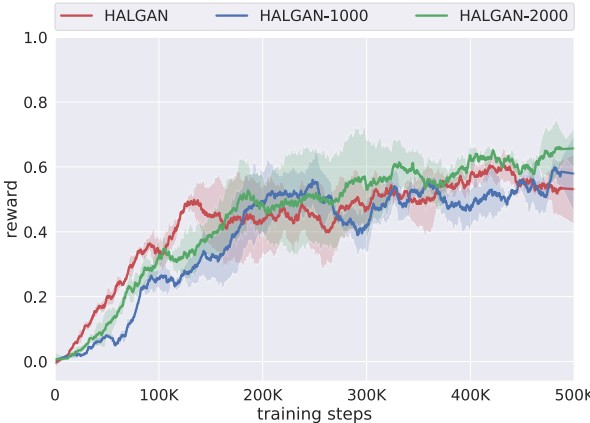

*Figure 6.* Reinforcement learning using VHER in TurtleBot task with varying size of training dataset for HALGAN. The curves being similar is a positive result that shows only minor variance of RL agent performance with training data available for HALGAN from 6800 (original) down to 1000 *near goal* training samples.

ifest themselves acutely when one attempts running the reinforcement learning process in the real world.

In this work, we have shown that Hindsight Experience Replay can be extended to visual scenarios by retroactively hallucinating goals into agent observations. We empirically prove that by utilizing failed trajectories in such a way, the agent can begin learning to solve tasks immediately. HALGAN+HER trained agent converges faster than standard RL techniques on two navigation tasks in a 3D environment and a simulated robotics application. HALGAN requires relatively few snapshots of *near goal* images with known goal configurations, in contrast to standard HER which assumes knowledge of goal location throughout training. In certain environments, this dataset could be generated online as the agent learns, or supplied from orthogonal techniques such as GoalGAN (Held et al., 2017). We leave this as an avenue for future work.

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

# Appendix

## A. Experimental Hyperparameters

Refer to table below for environment specific hyperparameters.

| HYPERPARAMETER | TURTLEBOT | MINIWORLD NAVIGATE | MINIWORLD PICK-AND-PLACE |
|---|---|---|---|
| REPLAY WARMUP | 10,000 | 10,000 | 10,000 |
| REPLAY CAPACITY | 100,000 | 100,000 | 100,000 |
| INITIAL EXPLORATION $\epsilon$ | 1.0 | 1.0 | 1.0 |
| FINAL EXPLORATION $\epsilon$ | 0.5 | 0.5 | 0.5 |
| $\epsilon$ ANNEAL STEPS | 100,000 | 100,000 | 250,000 |
| DISCOUNT ($\gamma$) | 0.99 | 0.99 | 0.99 |
| OFF-POLICY ALGORITHM | DDQN | DDPG | DDPG |
| POLICY OPTIMIZER | ADAM | ADAM | ADAM |
| LEARNING RATE | $1e^{-3}$ | $1e^{-5}$ (ACTOR), $1e^{-4}$ (CRITIC) | $1e^{-5}$ (ACTOR), $1e^{-4}$ (CRITIC) |
| SIZE OF $R$ FOR HALGAN | 6,840 | 2,000 | 6,419 |
| HALLUCINATION START % | 20% | 30% | 30% |
| HALLUCINATION END % | 0% | 0% | 0% |
| MAX FAILED TRAJECTORY LENGTH | 16 | 32 | 16 |
| IMAGE SIZE | 64x64 | 64x64 | 64x64 |
| RANDOM SEEDS | 75839, 69045, 47040 | 75839, 69045, 47040, 60489, 11798 | 75839, 69045, 47040, 60489, 11798 |

*Table 1.* Environment Specific Hyperparameters

Refer to table below for HALGAN specific hyperparameters.

| HYPERPARAMETER | VALUE |
|---|---|
| LATENT VECTOR SIZE | 128 |
| LATENT SAMPLING DISTRIBUTION | $\mathcal{N}(1, 0.1)$ |
| AUXILIARY TASK WEIGHT | 10 |
| GRADIENT PENALTY WEIGHT | 10 |
| $L_2$ LOSS ON $H$ WEIGHT | 1 |
| OPTIMIZER | ADAM |
| LEARNING RATE | $1e-4$ |
| ADAM $\beta1$ | 0.5 |
| ADAM $\beta2$ | 0.9 |
| $D$ ITERS PER $H$ ITER | 5 |

*Table 2.* Hyperparameters involved in training HALGAN

## B. Network Architectures

Refer to table below for details on the network architecture for DDQN. LeakyReLu's were used as activations throughout except for the output layer where no activation was used.

| LAYER | SHAPE | FILTERS | #PARAMS |
|---|---|---|---|
| IMAGE INPUT | 64x64 | 3 | 0 |
| CONV 1 | 5x5 | 4 | 304 |
| CONV 2 | 5x5 | 8 | 808 |
| CONV 3 | 5x5 | 16 | 3216 |
| CONV 4 | 5x5 | 32 | 12832 |
| DENSE 1 | 32 | - | 16416 |
| DENSE 2 | 4 (*nbactions*) | - | 132 |
| TOTAL | - | - | 33708 |

*Table 3.* Network Architecture for DDQN Agent

Refer to table below for details on the network architecture for actor for DDPG. LeakyReLu's were used as activations throughout except for the output layer where a Tanh was used.

| LAYER | SHAPE | FILTERS | #PARAMS |
|---|---|---|---|
| IMAGE INPUT | 64x64 | 3 | 0 |
| CONV 1 | 5x5 | 4 | 304 |
| CONV 2 | 5x5 | 8 | 808 |
| CONV 3 | 5x5 | 16 | 3216 |
| CONV 4 | 5x5 | 32 | 12832 |
| DENSE 1 | 32 | - | 16416 |
| DENSE 2 | 2 (*nbactions*) | - | 66 |
| TOTAL | - | - | 33642 |

*Table 4.* Network Architecture for DDPG Actor

Refer to table below for details on the network architecture for critic for DDPG. LeakyReLu's were used as activations throughout except for the output layer where no activation was used.

| LAYER | SHAPE | FILTERS | #PARAMS |
|---|---|---|---|
| IMAGE INPUT | 64x64 | 3 | 0 |
| CONV 1 | 5x5 | 4 | 304 |
| CONV 2 | 5x5 | 8 | 808 |
| CONV 3 | 5x5 | 16 | 3216 |
| CONV 4 | 5x5 | 32 | 12832 |
| DENSE 1 | 32 | - | 16416 |
| DENSE 2 | 1 | - | 33 |
| TOTAL | - | - | 33673 |

*Table 5.* Network Architecture for DDPG Critic

Refer to table below for details on the network architecture for the generator in HALGAN. LeakyReLu's were used as

activations throughout except immediately after the conditioning layer where no activation was used and the output where tanh was used.

| LAYER | SHAPE | FILTERS | #PARAMS |
|---|---|---|---|
| CONFIG INPUT | 3 | - | 0 |
| DENSE 1 | 128 | - | 384 |
| CONDITIONING INPUT | 128 | - | 0 |
| MULTIPLY | 128 | - | 0 |
| RESHAPE | 1x1 | 128 | 0 |
| UPSAMPLE + CONV 1 | 4x4 | 64 | 131136 |
| BATCHNORM | 2x2 | 64 | 256 |
| UPSAMPLE + CONV 2 | 4x4 | 64 | 65600 |
| BATCHNORM | 4x4 | 64 | 256 |
| UPSAMPLE + CONV 3 | 4x4 | 64 | 65600 |
| BATCHNORM | 8x8 | 64 | 256 |
| UPSAMPLE + CONV 4 | 4x4 | 32 | 32800 |
| BATCHNORM | 16x16 | 32 | 256 |
| UPSAMPLE + CONV 5 | 4x4 | 32 | 16416 |
| BATCHNORM | 32x32 | 32 | 128 |
| UPSAMPLE + CONV 6 | 4x4 | 16 | 8028 |
| BATCHNORM | 64x64 | 16 | 64 |
| CONV 7 | 4x4 | 8 | 2056 |
| BATCHNORM | 64x64 | 8 | 32 |
| CONV 8 | 4x4 | 3 | 387 |
| TOTAL | - | - | 323707 |

*Table 6.* Network Architecture HALGAN Generator

Refer to table below for details on the network architecture for the discriminator in HALGAN. LeakyReLu's were used as activations throughout except at the output where no activation was used.

| LAYER | SHAPE | FILTERS | #PARAMS |
|---|---|---|---|
| IMAGE INPUT | 64x64 | 3 | 0 |
| CONV 1 | 4x4 | 32 | 1568 |
| CONV 2 | 4x4 | 32 | 16416 |
| CONV 3 | 4x4 | 32 | 16416 |
| CONV 4 | 4x4 | 64 | 32832 |
| CONV 5 | 4x4 | 64 | 65600 |
| CONV 6 | 4x4 | 64 | 65600 |
| CONV 7 | 4x4 | 128 | 131200 |
| DENSE (AUX) | 2 | - | 129 |
| DENSE (REAL/FAKE) | 1: | - | 258 |
| TOTAL | - | - | 330019 |

*Table 7.* Network Architecture for HALGAN Discriminator