# OpenReview forum: "Addressing Sample Complexity in Visual Tasks Using Hindsight Experience Replay and Hallucinatory GANs"
_ICML.cc/2019/Workshop/RL4RealLife — RL4RealLife 2019_

### Official Review · AnonReviewer1 · 2019-05-22
**GANs to add targets to failed trajectories for HER in visual tasks**

**Rating:** 4
**Confidence:** 4

**Review:**

This paper presents an approach to applying HER in visual goal tasks by using GANs. If the agent's task is to navigate to a particular goal object, or move a particular object, the object must be in the scene at the appropriate location in the failed trajectories for HER to work. They train a GAN to add the target onto the scene in the appropriate location so that these trajectories can be added to the agent's replay buffer as in HER. They demonstrate results across 3 tasks showing the benefits of this approach.

The GAN makes use of the relative configuration of the robot to the desired goal state to generate the target at the right location in the image. The authors suggest this is reasonable because the agent could be doing SLAM or some other state tracking technique. I wonder 1) how dependent the method is on the relative configuration being accurate; and 2) whether this extra information could be used in some simpler way than hallucinating targets in the trajectory images.

VHER is also given demonstrations of the agent reaching the real target task to train the GAN. How much does these demonstrations help the faster learning of VHER compared to HER and DDQN? If you add those same demonstrations to their initial replay buffers, do they similarly learn faster?

Clarity:
The initial motivation for the paper compared to Nair et al could be made much clearer. It's not initially clear why targets may need to be added to entire failed trajectories rather than just conditioning on the image of the final state.

Originality:
The paper is very original, and presents an interesting use of GANS to modify failed trajectories to use within HER in visual tasks.

Significance:
This approach can significantly improve learning speed on visual tasks of these types.

Pros:
- Novel approach combining GANs into HER for visual tasks
- Nice results compared to basic HER, other variations, and DDPG/DDQN.
- Sample efficiency gains may make it reasonable to learn on real tasks and robots.

Cons:
- Motivation for adding targets to trajectories could be made clearer.
- Some questions around the fairness of the comparisons with VHER having relative configurations and demonstrations.

---

### Official Review · AnonReviewer2 · 2019-05-28
**An extension of HER to incorporate visual tasks; Combination with WGANs for learning reward mapping**

**Rating:** 3
**Confidence:** 4

**Review:**

In this work, the authors proposed HALGAN, a technique that extends Hindsight Experience Replay (HER) to tackle goal-driven visual navigation tasks. While HER has the issue of assuming the knowledge of goal mapping function f_g, which is not available in many visual-based tasks, in the proposed HALGAN framework the authors combine techniques from Wasserstein GANs to jointly learn such a goal mapping function (for altering the states of a failed trajectory into ones with near-goal states), together with RL policy.

In general I find this paper proposes an interesting idea of combining WGANs with the goal-generative idea of HER (in slightly modifying failed trajectories to generate good ones). My major comments/questions are: 1) the novelty of this work seems incremental, and the idea seems to be an extension of HER; 2) The empirical experiments on several benchmarks (and comparisons with VAE) is reasonable, but it'd be more comprehensive to understand if HALGAN can be compared/combined with similar ideas such as GoalGAN, in terms of better addressing sample complexity.

---

### Decision · Program_Chairs · 2019-05-28

Accept